# Serum Transforming Growth Factor β1 and Its Genetic Variants Are Associated with Increased Macrophage Inflammatory Protein 1β and Susceptibility to Idiopathic Carpal Tunnel Syndrome

**DOI:** 10.3390/jpm13050715

**Published:** 2023-04-24

**Authors:** Shaimaa A. Fattah, Mohamed S. Selim, Maha A. Abdel Fattah, Dina M. Abo-Elmatty, Noha M. Mesbah, Asmaa R. Abdel-hamed

**Affiliations:** 1Department of Biochemistry, Faculty of Pharmacy, Suez Canal University, Ismailia 41522, Egyptnoha_mesbah@pharm.suez.edu.eg (N.M.M.); asmaa.ramdan@pharm.suez.edu.eg (A.R.A.-h.); 2Cardiovascular Therapy Department, Novartis Company, Riyadh 12271, Saudi Arabia; mohamed.sleem@novartis.com; 3Department of Physical Medicine, Rheumatology and Rehabilitation, Faculty of Medicine, Suez Canal University, Ismailia 41522, Egypt; dr.mahaabdelfattah86@gmail.com

**Keywords:** transforming growth factors, macrophage inflammatory protein-1, single nucleotide polymorphism, carpal tunnel syndrome

## Abstract

Carpal tunnel syndrome (CTS) is a common entrapment neuropathy in which one of the body’s peripheral nerves becomes pinched or crushed. Transforming growth factor beta 1 (TGF-β1) plays an important role in the pathogenesis of CTS. An association between TGF-β1 polymorphisms and the susceptibility or progression of a number of diseases has been reported. In this study, three TGF-β1 single nucleotide polymorphisms (SNPs), serum TGF-β1, and macrophage inflammatory protein 1 beta (MIP-1β) were investigated as potential diagnostic markers for the progression of CTS in Egyptian patients. One hundred CTS patients and 100 healthy controls were recruited for the study. TGF-β1 SNPs +915G/C, −509C/T and −800G/A were determined by TaqMan genotyping assay. Serum TGF-β1 and MIP-1β levels were measured by ELISA. Serum TGF-β1 and MIP-1β levels increased significantly and were strongly correlated with the occurrence of CTS. The C allele of +915G/C, the T allele of −509C/T, and the G allele of −800G/A occurred more frequently in patients from CTS than in controls. The serum levels of TGF-β1 and MIP-1β in the group of carriers of the genotypes +915G/C GC and CC, the genotype −509C/T TT and the genotype −800G/A GA and AA were significantly higher in CTS patients. TGF-β1 and its +915G/C, −509C/T, and −800G/A SNPs and MIP-1β could be useful prognostic markers for the occurrence of CTS.

## 1. Introduction

Carpal tunnel syndrome (CTS) is a disabling condition that rheumatologists and orthopedic hand surgeons frequently deal with. It is a compression neuropathy defined as a mononeuropathy caused by a mechanical distortion due to a compressive force [1]. CTS affects 3% of the general population. CTS diagnosis occurs in every one in five individuals complaining of pain, numbness, and tingling in the hands. A CTS diagnosis is based on clinical examination and electrophysiological tests [2], with idiopathic CTS being the most common form [3]. Diagnostic criteria for CTS are clear. However, there are still many unknowns regarding CTS, including a possible genetic contribution to this disease and the molecular mechanisms that could help in diagnosis and treatment.

CTS is characterized by fibrosis of the subsynovial connective tissue (SSCT) surrounding the flexor tendons and median nerve in the carpal tunnel comprising all tissues between the visceral synovial sheath and the flexor tendons [4]. In addition, there are changes in the connective tissue of the median nerve: axonal degeneration, attraction and activation of macrophages, the release of inflammatory cytokines and nitric oxide, and development of chemical neuritis, all of which are consequences of compression [5]. Transforming growth factor beta (TGF-β), a pleiotropic growth factor, is involved in fibrosis of SSCT and can upregulate both fibroblast proliferation and activation [4].

Fibroblasts are an important source of constitutive and cytokine-induced C-C and C-X-C chemokines, such as MCP-1, macrophage inflammatory protein (MIP)-1 (MIP-1α, MIP-1β), RANTES, and IP-10 [6]. MIP-1β is an important regulator of fibrosis and is being investigated as a potential target for antifibrotic drugs [7]. The MIP-1β has been associated with the etiology of several diseases, including psoriasis vulgaris [8], sarcoidosis [9], cystic fibrosis [10], and multiple sclerosis [11]. MIP-1β and its corresponding receptors CCR1 and CCR5, which are regulated upon activation, are routinely upregulated in models of liver fibrogenesis [12]. Patients with pulmonary fibrosis (PF) also have elevated MIP-1β levels [13]. In addition to fibrosis caused by MIP-1, MIP-1 activation of CCR1, CCR3, and CCR5 leads to Ca^2+^ release, the elevation of activation indicators, and the production of proinflammatory mediators such as LTC4, arachidonic acid, and histamine [14].

In neurons, chemokines are suspected to modulate neuronal excitability and contribute to neuropathic pain [15]. The increased expression of chemokines in CTS patients suggests that they play a role in the recruitment of leukocytes to inflamed tissue and possibly in the direct or indirect mediation of neuropathic pain symptoms [16]. In CTS patients, the chemokine MIP-1β was easily distinguished from controls [16].

TGF-β1 treatment stimulated macrophage migration and induction of MIP-1α via the GTP-RhoA/NF-kB/AP-1 axis [17]. In contrast, further studies showed that TGF-β1 spared the MIP-1β response [18]. In general, cell activation is often a prerequisite for the production (and release) of MIP-1α and MIP-1β. These TGF-β1 mediated signaling pathways are potential targets for the treatment of CTS. However, the link between TGF-β1 and MIP-1β in CTS has not yet been investigated. We hypothesize that increased TGF-β1 levels are associated with increased serum MIP-1β levels, which in turn induces inflammation and fibrosis in CTS patients.

Although increased TGF-β1 gene expression has been studied in CTS patients [4,19], little is known about its etiology. Therefore, we propose that possible genetic TGF-β1 variants may be responsible for the elevation of serum TGF-β1 levels, which in turn leads to the development of CTS. TGF-β is encoded by three different genes: TGF-β1, TGF-β2, and TGF-β3. TGF-β1 is the most abundantly expressed isoform and has been associated with various disease susceptibilities [20]. Human TGF-β1 is located on chromosome 19q13.1 and is divided into seven exons, and the differences in TGF-β1 production in different people are thought to be genetically determined [21]. This study aimed to determine the association of TGF-β1 gene polymorphisms [rs1800471 (+915G/C), rs1800469 (−509C/T), and rs1800468 (−800G/A)] with elevated serum TGF-β1 levels and concomitant elevation of serum MIP-1β and susceptibility to CTS.

## 2. Materials and Methods

This case-control study included 100 CTS patients and 100 apparently healthy control subjects matched for age and sex. The control group had no history of CTS or other chronic diseases such as diabetes mellitus (DM), thyroid disease, using contraceptive pills, pregnancy, and rheumatoid arthritis (RA). Population stratification was unlikely because all participants were of the same ethnic group. The control group and CTS patients were matched for type of occupation and duration of exposure to wrist activities. With a power of 80% and a significance level of *p* < 0.05, the algorithm of Charan and Biswas [22] was used to calculate the sample size required to demonstrate the association between SNPs and CTS.

CTS patients were referred from the rheumatology and orthopedic outpatient clinics of Suez Canal University Hospital, Ismailia, Egypt, between September 2020 and September 2021. Diagnosis of CTS was based on the positivity of Tinel’s sign, the positivity of Phalen’s maneuver, disease sites, electromyography findings, and clinical stage. Patients’ age, sex, duration of symptoms, and dominant hand were determined. Weight and height were determined by anthropometric examination, from which body mass index (BMI) was calculated.

Patients with any of the following conditions were excluded from participation in the study: history of systemic diseases causing CTS, such as diabetes mellitus or hypothyroidism; sensory or motor deficits of the ulnar nerve after previous surgery CTS; multiple diagnoses of the upper extremities, such as lateral epicondylitis or cervical radiculopathy; concurrent systemic musculoskeletal diseases, such as RA or fibromyalgia; pregnancy.

### 2.1. Lab Analysis

Six milliliters of peripheral blood were drawn from each patient after a 12-h fast; two milliliters were collected in EDTA for DNA extraction and four milliliters were collected in plain tubes for serum separation. TGF-β1 and MIP-1β in serum were measured using the human-specific enzyme-linked immunosorbent assay (ELISA) (Cat. No. MBS590024, MBS824550, BioSource International, San Diego, CA, USA).

### 2.2. Genotyping

Genomic DNA was extracted from 300 μL of venous blood using the Wizard^®^ Genomic DNA Purification Kit (Promega, Madison, WI, USA). A NanoDrop ND-1000 (NanoDrop Tech., Inc., Wilmington, DE, USA) spectrophotometer was used to quantify and evaluate the purity of the isolated DNA. TGF-β1 +915G > C rs1800471, −509C > T rs1800469, and −800G > A rs1800468 were genotyped using real-time polymerase chain reaction (RT-PCR) allelic discrimination technology (probe ID: C__11464118_30, C___8708473_10, C___8708474_20, respectively; Applied Biosystem, Foster City, CA, USA). The reaction volume was 20 μL, including 10 μL Taqman Universal PCR Master Mix, No AmpErase UNG (2×), 20 ng genomic DNA diluted to 9.5 μL with DNase-RNase-free water, and 0.5 μL 20× TaqMan SNP Genotyping Assay Mix. Reactions were duplicated and subjected to proper controls in each run. Cycling conditions were: denaturation at 95 °C for 10 min, followed by 40 cycles of denaturation at 95 °C for 15 s and annealing and extension at 60 °C for 1 min. The analyzer AB 7500HT with the software Sequence Detection System (SDS) version 2.1.1 (Applied Biosystems, Cairo, Egypt) was used for PCR reactions. Genotyping was performed blindly for case/control status. Separate runs were performed for re-genotyping 10% of the randomly selected samples to eliminate the possibility of incorrect genotyping with a concordance rate of 100%.

### 2.3. Statistical Analysis

All statistical analyses were carried out using Statistical Package for Social Sciences (SPSS) software, version 17 (Chicago, IL, USA). For quantitative variables, descriptive statistics were represented as mean ± standard deviation (SD). Fisher’s exact test was used to compare the distribution of genotypes and allele frequencies. A 95% confidence interval (CI) and odds ratios (OR) were calculated. The chi-square test (χ2) was used to examine the Weinberg–Hardy equilibrium. Relationships between genotypes and laboratory parameters in CTS patients were examined using an independent *t*-test and a one-way (ANOVA) test with Tukey’s posthoc for multiple comparisons. Pearson’s correlation coefficient (Pearson’s r) with Bonferroni post hoc test was performed to correct for multiple comparisons. Logistic regression was performed to evaluate the association between the potential variables and CTS. Receiver operating characteristic curves (ROC) were used to evaluate the effectiveness of the variables as diagnostic indicators. All *p* values are two-tailed, and statistical significance was set at *p* < 0.05. Interventionary studies involving animals or humans, and other studies that require ethical approval, must list the authority that provided approval and the corresponding ethical approval code.

## 3. Results

### 3.1. Clinical and Demographic Characteristics of the Research Population

The demographic and clinical data of the CTS patients and the age- and sex-matched control group are summarized in Table 1. Compared with the control group, there were no significant differences in BMI and family history. The majority of patients had bilateral affected hands. Dominant hands were affected in 59% of patients and nondominant hands in 41%. Regarding electromyographic findings, 8% of patients had severe CTS, 38% moderate CTS, and 53% mild CTS.

### 3.2. Relationship between Serum TGF-β1 and MIP-1β and Susceptibility to Idiopathic CTS

There was a significant increase in serum TGF-β1 and MIP-1β in CTS patients compared with healthy controls (mean ± SD, 142.2 ± 20.6, 102.4 ± 26.8, *p* < 0.001—137.7 ± 18.6, 82.3 ± 20.0, *p* < 0.001; respectively) (Figure 1a). Serum TGF-β1 showed a strong positive correlation with serum MIP-1β levels in CTS patients (Figure 1b).

Based on a multivariate logistic analysis, TGF-β1 and MIP-1β were effective predictors of idiopathic CTS (Table 2). To test whether TGF-β1 and MIP-1β could be used as diagnostic biomarkers for idiopathic CTS, sensitivity and specificity were measured by ROC curve analysis. The area under the ROC curve (95% CI) of TGF-β1 and MIP-1β to discriminate CTS patients was [0.85 (0.8–0.9) *p* < 0.001, 0.96 (0.93–0.98) *p* < 0.001, respectively]. At the optimal cut-off value of 72 pG/mL TGF-β1, sensitivity was 86% and specificity was 100%. At the optimal cut-off value of 64.2 pG/mL MIP-1β, sensitivity was 82% and specificity was 100% (Figure 2).

### 3.3. Genotypes and Allele Frequencies

#### 3.3.1. TGF-β1: +915G > C rs1800471 Single Nucleotide Polymorphism

The entire study population was included in the study. The distribution of all +915G > C rs1800471 genetic variants did not differ from the prediction of Hardy–Weinberg equilibrium (*p* > 0.05). As shown in Table 3, the minor high-risk C allele for +915G > C rs1800471 was more frequent in the CTS patient group (16%) than in the control group (9%) [OR (95% CI)] [0.5 (0.3–0.9)]. In the high-risk group, 26% of patients were homozygous and heterozygous genotype carriers (GC + CC) of the C allele, compared with 13% in the control group [0.4 (0.2–0.9)]. These results suggest that the minor C allele may be associated with an increased risk of CTS.

When the association between the +915G > C rs1800471 polymorphism and laboratory variables was examined, carriers of the GC + CC genotype at high risk showed a significant increase in serum levels of TGF-β1 and MIP-1β compared with carriers of the GG genotype at low risk (Table 3).

#### 3.3.2. TGF-β1: −509C > T rs1800469 Single Nucleotide Polymorphism

SNP analysis of −509C > T rs1800469 showed that the minor T allele was more frequent in the patient group (32.5%,65 patients) than in the healthy control group (20.5%,41 patients) [OR (95%CI)] [0.5 (0.4–0.8)]. In the high-risk group, 17% were homozygous carriers of allele (T) of the patients and 8% of the control group, which showed an increased likelihood of CTS compared with CC [0.4 (0.1–0.9)]. In the low-risk group, homozygous carriers of allele (C) accounted for 67% of the control group and 52% of the patient group (Table 4), which was associated with a lower probability of CTS. The distribution of all genotypes was consistent with Hardy–Weinberg equilibrium in the study population (*p* > 0.05).

Regarding the association between 509C > T rs1800469 and the laboratory variables studied, carriers of the high-risk CT and TT genotypes had higher serum TGF-β1 and MIP-1β levels than those with the CC allele (Table 4).

#### 3.3.3. TGF-β1: −800G > A rs1800468 Single Nucleotide Polymorphism

As shown in Table 5, the incidence of −800G > A Major G allele was significantly higher in the patient group (73.5%) than in the control group (62%) [OR (95%CI)] [1.7 (1.1–2.6)]. Similarly, carriers of the homozygous GG genotype were associated with a higher probability of CTS than carriers of the GA and AA genotypes [2.7 (1.1–6.2)].

Laboratory studies revealed that serum levels of TGF-β1 and MIP-1β were significantly elevated in carriers of the G allele (GG and GA) compared with carriers of the AA genotype (Table 5).

## 4. Discussion

Because the symptoms of CTS do not respond to treatment with wrist splints, nonsteroidal anti-inflammatory drugs (NSAIDs), and corticosteroids for a long time, carpal tunnel decompression surgery is recommended in CTS patients, especially in severe cases [23]. Therefore, new ideas for therapy of CTS symptoms instead of surgery are essential.

CTS is characterized by an increase in TGF-β1 which is involved in fibrosis of SSCT and fibroblast proliferation [4]. Our data is consistent with that of previous studies that reported increased TGF-β1 in idiopathic CTS [4,24]. The use of TGF-β type I receptor (TGF-βRI) inhibitor SB431542 and TGF-β1 siRNA as well as TGF-β1 antagonist Smad7 decreased TGF-β1-induced inflammation in neurons [25]. Thus, inhibition of TGF-β1 to counteract the symptoms is a promising therapy for CTS.

Results from this study showed that CTS patients had a significant increase in serum MIP-1β levels compared with the control group, which may contribute to neuropathic symptoms. These results are consistent with Gila Moalem-Taylor et al. [16], who found a remarkable increase in MIP-1β in CTS patients compared to control subjects. Previous studies have shown that TGF-β1 induces the production of IL-1β in corneal fibrosis [26,27]. In some neurodegenerative diseases, TGF-β1 stimulates the production of inflammatory cytokines such as IL-1β and TNF-via phosphorylation of Smad proteins, including Smad2 and Smad3, in rat neurons [25]. In addition, Ishinaga et al. [28] showed that NF-kappaB-dependent transcription of IL-1β is regulated by TGF-β1 signaling. Furthermore, both immune and non-immune cells can express chemokines (MIP-1β) in response to IL-1β via up-regulation of TLR4 and its downstream targets MyD88 and TRAF6, leading to activation of NF-кB [29,30,31]. Results from this study agree with these data, revealing a strong positive correlation between TGF-β1 and MIP-1β. TGF-β1 and MIP-1β are promising prognostic biomarkers and therapeutic targets for CTS.

To date, several polymorphisms have been identified in the TGF-β1 gene, including rs1800471 (+915G/C), rs1800469 (−509C/T), and rs1800468 (−800G/A). The polymorphism +915G/C is located in exon 1, −509C > T is located in the first negative regulatory region, and −800G > A, is located in enhancer region 1. Individuals with the TT genotype have increased expression of TGF-β1 compared with individuals with the CC genotype. The affinity of the cAMP response element binding protein (CREB) family is reduced in the presence of allele A, which is associated with reduced TGF-β1 levels [20].

Many studies have investigated the association of TGF1β-SNPs with various diseases [32,33,34], but these SNPs have not yet been studied in CTS patients. We hypothesized that the increase in serum TGF-β1 in CTS patients might be due to genetic variants.

Based on this study, the rs1800471 C allele was associated with the risk of CTS. These results are in line with reports on RA, where rs1800471 C > G influenced the inflammatory environment in RA [34]. Furthermore, Nabrdalik et al. [32], showed that the minor C allele of rs1800471 was associated with the occurrence of chronic kidney disease, whereas the rs1800471 G allele was associated with the presence of hypertension [32]. Our results are consistent with previous studies that found an association between rs1800471 C > G and higher TGF-β1 levels and susceptibility to sepsis [33]. Consistent with these results, CTS patients carrying the minor C allele rs1800471 as either homozygotes CC or heterozygotes CG showed a significant increase in serum TGF-β1 levels compared with GG carriers. In contrast, these results are in contrast to Kiliś-Pstrusińska et al. [35], who found no association between rs1800471 C > G SNPs and TGF-β1. Previous studies have shown that the TGF-β1-associated SNP rs1800471 C > G influences NF-KB-associated inflammatory signaling in breast cancer susceptibility [36,37]. Since NF-KB is involved in the induction and release of MIP-1β via the TLR4/MyD88 pathway, the TGF-β1-associated SNP rs1800471 C > G might affect the induction of MIP-1β. Consistent with this assumption, our result showed that CTS patients carrying the minor C allele of rs1800471 had significantly higher serum MIP-1β levels than patients carrying the major G allele.

TGF-β1 rs1800469 (−509 T > C) has been associated with various diseases, such as susceptibility to atherosclerotic cerebral infarction [38], and end-stage renal disease [39]. Our results also showed that rs1800469 T > C was more prevalent in CTS patients than in the control group. In addition, the relative increase of serum TGF-β1 was significantly higher in carriers of the minor T allele as homozygous TT or heterozygous CT than in CTS patients with major C allele, indicating a possible functional role. Since TGF-β1 is a pleiotropic factor, previous studies have shown that decreased circulating TGF-β1 due to the rs1800469 CC genotype was associated with tumor progression and susceptibility [40], and myopia in the European population [41]. CTS patients carrying the TT genotype showed greatly increased serum MIP-1β levels than those with CT.

Regarding rs1800468 (−800G/A), CTS patients had an increased frequency of major allele G. It is controversial that minor allele A > G is associated with the occurrence of cervical cancer [42] and coronary heart disease [43]. In addition, our results showed that the minor protective A allele (GA and AA) resulted in a significant increase in serum TGF-β1 and MIP-1β levels compared with those carrying the major risk allele, G. Bogacz et al. [44], reported that renal transplant patients with the A allele rs1800468 required a slightly lower dose of the immunosuppressant tacrolimus-TAC than patients with the G allele. Therefore, the TGF-β1 SNPs are potential targets for CTS therapy.

This study is limited by a relatively small sample size. Furthermore, our results are limited to the Egyptian population in the Suez Canal area. Therefore, further studies should be performed in other populations to confirm that TGF-β1 rs1800469 (−509T > C) and rs1800468 (−800G > A) SNPs are potential CTS susceptibility markers. Moreover, further studies are recommended to determine and elucidate the mechanism of TGF-β1-induced MIP-1β release in CTS patients. We recommend further studies to confirm the potential of a TGF-β1 inhibition treatment-based regimen as an adjuvant therapy for CTS.

## 5. Conclusions

By way of conclusion, the current study provides a new insight into the correlation between TGF-β1 and MIP-1β in CTS patients. Increased serum TGF-β1 and MIP-1β occurred in CTS patients. This study demonstrated for the first time the association of the allelic variants of the TGF-β1 gene rs1800471 (+915C > G), rs1800469 (−509T > C) and rs1800468 (−800G > A) SNPs with CTS, in which significantly increased serum levels of TGF-β1 and MIP-1β were found. Accordingly, we hypothesized that TGF-β1 and its SNPs may be involved in the induction and release of MIP-1β in CTS patients and may be a crucial risk factor for the pathogenetic susceptibility of CTS. This could contribute to advances in the diagnosis and treatment of CTS.

## Figures and Tables

**Figure 1 jpm-13-00715-f001:**
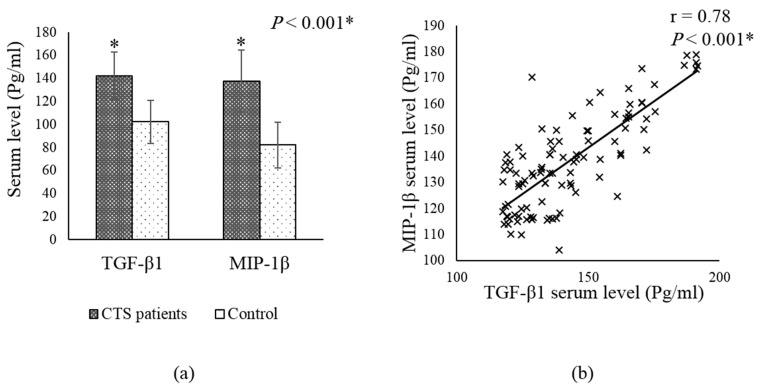
(**a**) TGF-β1 and MIP-1β in control and CTS patients group. Data are presented as mean ± SD. * Indicates a significant difference from the control group at *p* < 0.05. (**b**) Correlation in ZNF804a gene expression and TNF-α serum levels in RA group. r indicates Pearson’s correlation coefficient. CTS; carpal tunnel syndrome, TGF-β1; transforming growth factor β1, MIP-1β; macrophage inflammatory protein 1β.

**Figure 2 jpm-13-00715-f002:**
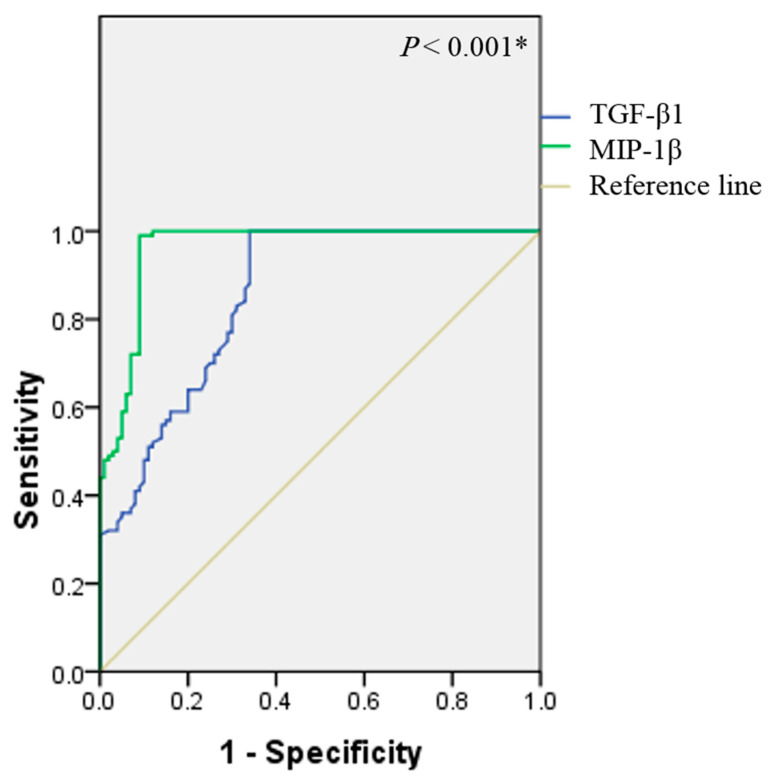
ROC curve for TGF-β1 and MIP-1β diagnosis assay in CTS patients. CTS; carpal tunnel syndrome, TGF-β1; transforming growth factor β1, MIP-1β; macrophage inflammatory protein 1β. * Indicates a significant difference from the control group at *p* < 0.05.

**Table 1 jpm-13-00715-t001:** Demographic and clinical characteristics of the study groups.

Variables	Control (*n* = 100)	CTS (*n* = 100)	*p* Value
Age	46.4 ± 11.8	47.5 ± 14	0.5
Sex			0.2
Male	40	30	
Female	60	70	
Duration of the disease (years)	-----	43 ± 11	-----
BMI (kg/m^2^)	29.4 ± 5.1	29.3 ± 5.6	0.9
Family history	-----	47	-----
Affected hand (side) (right/left/bilateral)	-----	25/26/49	-----
Dominant hand (dominant/non-dominant)	-----	59/41	-----
EMG findings	-----		-----
Mild		53	
Moderate		38	
severe		9	

Values are represented as mean ± SD. Comparisons were performed by student *t*-test.

**Table 2 jpm-13-00715-t002:** Logistic regression analysis for the evaluation of risk variables for the occurrence of carpal tunnel syndrome.

Variables	OR	95%CI	*p*-Value
TGF-β1 (Pg/mL)	1.09	1.0–1.1	<0.001 *
MIP-1β (Pg/mL)	1.13	1.1–1.2	<0.001 *

TGF-β1; transforming growth factor β1, MIP-1β; macrophage inflammatory protein 1β. OR; odds ratio, CI; confidence interval. * Indicates a significant difference at *p* < 0.05.

**Table 3 jpm-13-00715-t003:** Allele frequencies and genotype distribution of the TGF-B1: +915G > C rs1800471 polymorphism and its relationship to laboratory variables in the control group and CTS patients.

Variables		Controls/Patients (*n*)	*p*-Value
+915G > C alleles	G	183/168	0.03 * a
	C	17/32	
+915G > C genotypes	GG	87/74	0.03 * b
	GC	9/16	
	CC	4/10	
	GC + CC	13/26	
		Patients	
TGF-β1 (pg/mL)	GG	133.9 ± 13.5	<0.001 * b
	GC + CC	165.8 ± 19.1	
MIP-1β (pg/mL)	GG	131.8 ± 13.9	0.001 * b
	GC + CC	154.6 ± 20.2	

Allele frequencies and genotype distribution were performed by the Chi-Square test. Comparisons between genotype carriers and the laboratory variables were performed by one-way ANOVA test followed by Tukey’s test for multiple comparisons. Data are presented as mean ± SD. TGF-β1; transforming growth factor β1, MIP-1β; macrophage inflammatory protein 1β. * indicates a significant difference from CC genotype carriers at *p* < 0.05. a G VS. C. b CG + GG VS. CC.

**Table 4 jpm-13-00715-t004:** Allele frequencies and genotypes distribution of the TGF-B1: −509C > T rs1800469 polymorphism in the control group and CTS patients.

Variables		Controls/Patients (*n*)	*p*-Value
−509C > T alleles	C	159/135	0.008 * a
	T	41/65	
−509C > T genotypes	CC	67/52	
	CT	25/31	0.2 b
	TT	8/17	0.03 * c
		Patients	
TGF-β1 (pg/mL)	CC	134.7 ± 18.2	
	CT	145.2 ± 15.0	0.004 * b
	TT	159.7 ± 24.8	<0.001 * c
MIP-1β (pg/mL)	CC	131.9 ± 18.6	
	CT	139.9 ± 13.0	0.1 b
	TT	142.2 ± 20.6	<0.001 * c

Allele frequencies and genotype distribution were performed by the Chi-Square test. Comparisons between genotype carriers and the laboratory variables were performed by one-way ANOVA test followed by Tukey’s test for multiple comparisons. Data are presented as mean ± SD. TGF-β1; transforming growth factor β1, MIP-1β; macrophage inflammatory protein 1β. * indicates a significant difference from TT genotype carriers at *p* < 0.05. a C VS. T. b CT VS. TT. c. CC VS. TT.

**Table 5 jpm-13-00715-t005:** Allele frequencies and genotypes distribution of the TGF-B1: −800G > A rs1800468 polymorphism in the control group and CTS patients.

Variables		Controls/Patients (*n*)	*p*-Value
−800G > A alleles	G	124/147	0.02 * a
	A	76/53	
−800G > A genotypes	GG	45/57	
	GA	34/33	0.4 b
	AA	21/10	0.02 * c
		Patients	
TGF-β1 (pg/mL)	GG	149.5 ± 21.4	
	GA	134.8 ± 16.0	0.002 * b
	AA	125.1 ± 6.8	0.001 * c
MIP-1β (pg/mL)	GG	142.5 ± 18.8	
	GA	132.7 ± 17.8	0.04 * b
	AA	126.9 ± 11.3	0.03 * c

Allele frequencies and genotype distribution were performed by the Chi-Square test. Comparisons between genotype carriers and the laboratory variables were performed by one-way ANOVA test followed by Tukey’s test for multiple comparisons. Data are presented as mean ± SD. TGF-β1; transforming growth factor β1, MIP-1β; macrophage inflammatory protein 1β. * indicates a significant difference from GG genotype carriers at *p* < 0.05. a A vs. G. b GA VS. GG. c. AA VS. GG.

## Data Availability

Data is available within the article.

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
