# Peer review of "Serum Transforming Growth Factor β1 and Its Genetic Variants Are Associated with Increased Macrophage Inflammatory Protein 1β and Susceptibility to Idiopathic Carpal Tunnel Syndrome"

_jpm, 2023, doi:10.3390/jpm13050715_

Round 1
Reviewer 1 Report
Please, add an ethics section and report full name of the etihc research commintee that approved the study and numbre reference of approval, this is mandatory.
Please, explain methods used to control logistic regression assumptions and model selection variables (anova table, step backward, forward or both...)
Author Response
Dear DR,
Thank you for giving us the opportunity to submit the revised version of the manuscript with the code " JPM-2337134". We hope that it is now more suitable for publication in the " Journal of Personalized Medicine."
We thank the editor and reviewers for their careful review of our manuscript and their constructive comments. We have addressed the comments to improve and clarify the manuscript.
Reviewers` comments:
Reviewer #1:
Comments to authors
- Please, add an ethics section and report full name of the ethic research committee that approved the study and number reference of approval, this is mandatory.
Answer: We have already added the full name of the ethics committee that approved the study and the approval number in the Institutional Review Board Statement section before the references (Institutional Review Board Statement: The Ethics Committee of the Faculty of Pharmacy at Suez Canal University approved the study procedure (201811MH6), which was conducted in accordance with the principles of the Declaration of Helsinki (2000 revision)).
- Please, explain methods used to control logistic regression assumptions and model selection variables (anova table, step backward, forward or both...).
Answer: We performed multivariate analysis using the forward conditional method of stepwise binary logistic regression for TGF-1β and MIP-1β.
Reviewer 2 Report
Serum transforming growth factor beta 1 and its genetic variants are associated with increased macrophage inflammatory protein beta 1 and susceptibility to idiopathic carpal tunnel syndrome
This is an excellent article attempting to investigate TGF-1β single nucleotide polymorphisms (SNPs,) serum TGF-1β, and macrophage inflammatory 20 protein beta 1 (MIP-1β) as potential diagnostic markers for the progression of carpal tunnel syndrome (CTS) in Egyptian patients. The study finds that increased serum TGF-1β and MIP-1β levels were strongly correlated with the occurrence of CTS. It was also found that the C allele of +915G/C, the T allele of -509C/T, and the G allele of -800G/A occurred more frequently in patients from CTS compared to controls. Finally, it was concluded that TGF-1β and its +915G/C, -509C/T, and -800G/A SNPs and MIP-1β might serve as prognostic markers for the occurrence of CTS. The study is well-designed, experiments were performed meticulously, and the manuscript is well-written. However, the following are the specific comments to further strengthen the present manuscript,
1. The study was carried out using a small sample size of Egyptian patients, what is the global scenario?
2. Are there previous studies showing the involvement of genetic variations of TGF-1β in CTS that can be included in the discussion?
3. Does TGF-1β inhibition treatment could prevent CTS onset, or can this only be helpful to treat CTS?
Author Response
Dear DR,
Thank you for giving us the opportunity to submit the revised version of the manuscript with the code " JPM-2337134". We hope that it is now more suitable for publication in the " Journal of Personalized Medicine."
We thank the editor and reviewers for their careful review of our manuscript and their constructive comments. We have addressed the comments to improve and clarify the manuscript. The manuscript has been revised and modified according to the reviewers' suggestions.
Reviewer #2:
Comments to authors
This is an excellent article attempting to investigate TGF-1β single nucleotide polymorphisms (SNPs,) serum TGF-1β, and macrophage inflammatory 20 protein beta 1 (MIP-1β) as potential diagnostic markers for the progression of carpal tunnel syndrome (CTS) in Egyptian patients. The study finds that increased serum TGF-1β and MIP-1β levels were strongly correlated with the occurrence of CTS. It was also found that the C allele of +915G/C, the T allele of -509C/T, and the G allele of -800G/A occurred more frequently in patients from CTS compared to controls. Finally, it was concluded that TGF-1β and its +915G/C, -509C/T, and -800G/A SNPs and MIP-1β might serve as prognostic markers for the occurrence of CTS. The study is well-designed, experiments were performed meticulously, and the manuscript is well-written. However, the following are the specific comments to further strengthen the present manuscript,
- The study was carried out using a small sample size of Egyptian patients, what is the global scenario?
Answer: Thank you for your comments and valuable suggestions. We calculated the sample size according to the algorithm of Charan and Biswas (With a power of 80% and a significance level of p < 0.05, the algorithm of Charan and Biswas [22] was used to calculate the sample size required to demonstrate the association between SNPs and CTS). In addition, we added limitations and recommendations in the last paragraph before the conclusion that this study is limited by a relatively small sample size and the restriction to the Egyptian population in the Suez Canal area. Therefore, further studies should be performed in other populations to confirm that the TGF-1β rs1800469 (-509T > C) and rs1800468 (-800G > A) SNPs are potential CTS susceptibility markers.
- Are there previous studies showing the involvement of genetic variations of TGF-1β in CTS that can be included in the discussion?
Answer: There are previous studies showing the association between genetic variations of TGF-1β in atherosclerotic cerebral infarction, RA, hypertension, sepsis, cervical cancer, and end-stage renal disease, and we have already listed them in the Discussion. However, our study is the first study to show the association between these polymorphisms and the incidence of CTS.
- Does TGF-1β inhibition treatment could prevent CTS onset, or can this only be helpful to treat CTS?
Answer: Because the use of TGF-β type I receptor (TGF-βRI) inhibitor SB431542 and TGF-1β siRNA as well as TGF-1β antagonist Smad7 decreased TGF-1β-induced inflammation in neurons, inhibition of TGF-1β might counteract CTS symptoms as therapy. We also added in the recommendation that further studies are recommended to confirm the potential of a TGF-1β inhibition treatment-based regimen as adjuvant therapy for CTS.